# Awareness, knowledge, and beliefs about Human Papillomavirus and its vaccine among Egyptian medical students: A cross-sectional national study

Muhammad Elmanzlawey[1], Mohamed Terra[1]*, Mohamed Baklola[2]*, Yara Ahmed[3], Mariam Abuelela[4], Ibrahim Noureddin Al-Kurd[5], Abdel-Hady El-Gilany[6]

1 General Practitioner, Ministry of Health, Cairo, Egypt, 2 Faculty of Medicine, Mansoura University, Mansoura, Egypt, 3 Faculty of Medicine, Chuvash State University, Cheboksary, Russia, 4 Faculty of Medicine, Beni Suef University, Beni Suef, Egypt, 5 Faculty of Medicine, Benha University, Benha, Egypt, 6 Department of Public Health and Community Medicine, Faculty of Medicine, Mansoura University, Mansoura, Egypt

☯ These authors contributed equally to this work.
* Mohamedbaklola2000@gmail.com (MB); Mohamedtera75@gmail.com (MT)

## Abstract

### Background

Human papillomavirus (HPV) is a major cause of preventable cancers, including cervical, penile, and oropharyngeal cancers. Despite the availability of safe and effective vaccines, gaps in awareness, knowledge, and uptake persist globally, particularly in low- and middle-income countries. Medical students, as future healthcare providers, play a critical role in HPV prevention advocacy, making it important to understand their level of preparedness.

### Methods

A cross-sectional analytical study was conducted during the 2024–2025 academic year among medical students enrolled in faculties of medicine across Egypt. Eligible participants included undergraduate and internship-year students who voluntarily completed an online questionnaire adapted from the validated Karki et al. instrument, assessing HPV awareness, knowledge, beliefs, and vaccine uptake. The tool was reviewed by a panel of public health experts and pilot-tested among 30 students. Data were analyzed using descriptive statistics, chi-square tests, Mann–Whitney U tests, and multivariable logistic regression.

### Results

A total of 1,431 students participated; 91.0% were aware of HPV. Awareness was significantly associated with older age (p = 0.003), Egyptian nationality (p < 0.001),

**Data availability statement:** The data-sets generated and/or analyzed during this study are publicly available in the Figshare repository at https://doi.org/10.6084/m9.figshare.30417178.v1.

**Funding:** The author(s) received no specific funding for this work.

**Competing interests:** The authors have declared that no competing interests exist.

higher academic year (p < 0.001), urban residence (p = 0.002), and prior HPV-related education (p < 0.001). Knowledge of sexual transmission was high (97–98%), yet misconceptions persisted—67% believed HPV typically causes symptoms (p < 0.001), and 50% incorrectly identified HPV as herpesvirus (p = 0.004). Vaccine-related knowledge was limited: 78% believed the vaccine causes serious adverse effects (p < 0.001), and 70–76% thought it was intended only for sexually active individuals (p = 0.03). Logistic regression confirmed academic seniority (AOR = 2.41, 95% CI 1.85–3.12), urban residence (AOR = 1.74, 95% CI 1.28–2.36), and prior HPV education (AOR = 3.06, 95% CI 2.14–4.38) as strong independent predictors of awareness. Gender differences were also observed: males were more willing to vaccinate (44% vs. 36%, p = 0.02) but perceived lower personal risk (8% vs. 12%, p = 0.04), whereas females expressed greater safety concerns (22% vs. 15%, p = 0.03), higher perceived stigma (p = 0.01), and stronger recognition of HPV's cancer threat (95% vs. 97%, p = 0.04).

## Conclusion

While HPV awareness among Egyptian medical students is high, significant misconceptions about pathogenesis, prevention, and vaccine safety remain. Educational interventions, particularly those targeting early academic years, are essential to address these gaps, improve vaccine acceptance, and strengthen the role of future physicians in HPV prevention efforts.

## Introduction

Human papillomavirus (HPV) is one of the most widespread sexually transmitted infections globally and is a major cause of several cancers, including cervical, vulvar, anal, penile, and oropharyngeal cancers [1]. Two high-risk types, HPV 16 and HPV 18, are responsible for about 70% of cervical cancer cases worldwide [2]. Cervical cancer remains the second most common cancer among women, with an estimated 660,000 new cases and approximately 350,000 deaths annually [3,4]. Men are also significantly affected, with global data showing that nearly one in three men is infected with at least one genital HPV type, and around one in five carries one or more high-risk genotypes [5]. The burden of HPV-associated cancers is particularly high in low- and middle-income countries (LMICs), where they account for about 6.7% of all cancers, compared to 2.8% in high-income countries (HICs) [6].

Globally, HPV is the most common sexually transmitted infection, and nearly all sexually active individuals acquire it during their lifetime. The point prevalence of high-risk HPV types is approximately 11–12% among women, while genital HPV infection of any type is detected in about 45% of men, with roughly one quarter carrying high-risk genotypes [7]. HPV causes almost all cervical cancers and a substantial proportion of anal, vulvar, penile, and oropharyngeal cancers, with cervical cancer alone responsible for an estimated 660,000 new cases and 350,000 deaths annually, the majority occurring in low- and middle-income countries [8].

Oral HPV infection is also relatively common; population-based surveys report an overall oral HPV prevalence of around 7% in adults, with higher rates among men [9]. HPV16, the type most strongly linked to oropharyngeal cancer, is detected in approximately 1% of adults, with infection risk increasing with the number of oral sex partners and tobacco exposure [9]. Randomized trials and real-world data demonstrate that prophylactic vaccination reduces vaccine-type oral HPV infections. The Costa Rica Vaccine Trial showed approximately 83.5% efficacy against concurrent cervical, anal, and oral HPV16/18 detection among HPV-naïve women four years post-vaccination, while observational studies report 72–93% lower detection of vaccine-type oral HPV and markedly reduced odds of oral HPV16 among vaccinated individuals [10]. Collectively, these findings support the expectation that widespread vaccination will substantially reduce HPV-related oropharyngeal cancer burden over time, although continued surveillance is essential to quantify incidence reductions at the population level [11].

HPV vaccination is a proven, safe, and cost-effective method for preventing infection and related diseases in both sexes [12]. However, implementation varies widely, with coverage in LMICs reaching only about 40%, compared to over 80% in HICs [13]. Delayed introduction, slower rollout, and challenges such as affordability, competing health priorities, and limited resources contribute to this disparity [13]. While organizations like the World Health Organization (WHO) and Gavi, the Vaccine Alliance, have facilitated access in many settings, eligibility restrictions exclude some middle-income countries [14]. WHO guidelines recommend vaccinating girls aged 9–14, the period when the vaccine is most effective before the onset of sexual activity. Vaccinating boys in this age group is also important for reducing transmission and protecting high-risk populations [4]. Nonetheless, many individuals, particularly those aged 16 and older, remain unvaccinated. Furthermore, some countries focus vaccination efforts on females for herd immunity, overlooking at-risk male groups such as men who have sex with men (MSM) and heterosexual men with unvaccinated partners [13] Preventive screening for HPV-related cancers is also gender-skewed, with well-established cervical cancer screening protocols for women, whereas no validated or routinely implemented screening method exists for HPV-associated cancers in cisgender men. In addition, barriers and limitations persist for sexual and gender minority individuals assigned female at birth, as noted previously [15].

Research in the Middle East and North Africa (MENA) region has revealed considerable gaps in HPV awareness, knowledge, and vaccine uptake. In Egypt, one study among female university students found that only 2% had received the HPV vaccine and none had undergone Pap smear screening [16]. Knowledge levels were moderate, and many reported barriers such as not knowing where to access services, lack of perceived risk, and low self-efficacy. Another study among Egyptian medical students showed poor knowledge of HPV, especially regarding its link to oropharyngeal cancer, with only 7.7% aware of this association and less than one-third vaccinated [17]. The most cited barrier was lack of awareness, followed by cultural attitudes, limited accessibility, and high cost [17]. Similarly, a recent study in Saudi Arabia reported that just over half of university students had heard of HPV, and only 10% had been vaccinated [18]. Awareness of high-risk HPV types was low, and lack of education was the most common obstacle to vaccination.

These findings highlight a persistent gap in HPV-related knowledge and preventive practices among university students in the region, including those in health-related fields. Since medical students are future healthcare providers, their awareness and attitudes toward HPV prevention are crucial for effective public health messaging. This study aims to assess the levels of awareness, knowledge, and beliefs about HPV and the HPV vaccine among medical students in Egypt, and to identify factors associated with HPV awareness.

## Methods

### Study design and setting

This cross-sectional analytical study was conducted during the 2024–2025 academic year and targeted medical students enrolled in faculties of medicine across Egypt. Recruitment took place over a two-month period, from 01/10/2024–30/1/2025. The study cohort comprised Egyptian students registered in any medical institution.

## Participants and eligibility criteria

Eligible participants were undergraduate and internship-year medical students who voluntarily agreed to participate in the study. No restrictions were placed on age, gender, or academic year. Students who declined participation were not enrolled in the study.

## Sample size determination

The sample size was calculated using MedCalc version 15.8. The primary outcome of interest was the proportion of students aware of HPV. Based on a prior study among U.S. undergraduates reporting an awareness level of 91.5% [19], the following parameters were applied: alpha error of 5%, statistical power of 80%, and precision of 5%. The initial calculation yielded a required sample size of 125 students. After applying a design effect of 10, the final minimum required sample size was 1,250 students. Ultimately, 1,431 students were recruited, exceeding the required sample to enhance the study's statistical power.

## Data collection tool

The survey instrument used in this study was adopted from the validated questionnaire developed by Karki et al., 2021 in a study about HPV knowledge, beliefs, and vaccine uptake among United States and international college students [19]. The original tool comprises three structured scales in addition to socio-demographic and behavioral measures, and it has been previously validated for assessing HPV awareness, knowledge, beliefs, and vaccine uptake in university populations.

For the current study, the content was reviewed by a panel of three public health experts with experience in epidemiology and sexual health to ensure its cultural appropriateness and relevance for Egyptian medical students. Minor wording adjustments were made to enhance clarity without altering the meaning or scoring of any items. The questionnaire was then pilot tested among 30 medical students who were not included in the final sample. Feedback from the pilot phase was used to refine phrasing, confirm comprehension, and estimate completion time, which averaged 10–15 minutes.

The first section collected socio-demographic information, including age, gender, nationality, place of residence (urban or rural), and academic year. Sexual history was assessed through a single yes/no question: "Are you or have you been sexually active?".

HPV awareness was determined by the question, "Have you heard about HPV?" with "Yes" or "No" as response options. Those who answered "Yes" completed the HPV knowledge section, which contained 18 true/false items covering HPV transmission, clinical features, prevention, and associated health outcomes. Each correct answer was awarded one point, yielding a total score from 0 to 18, with higher scores indicating greater knowledge. The questionnaire did not include an item assessing the source of HPV awareness (e.g., academic curriculum, peers, or media exposure).

HPV vaccine awareness was similarly measured with a yes/no question: "Have you heard about the HPV vaccine?" Participants responding "Yes" proceeded to the HPV vaccine knowledge scale, which included eight true/false items assessing understanding of dosage, target population, safety, and effectiveness. Scores ranged from 0 to 8, with one point assigned per correct response.

HPV-related beliefs were assessed using the 12-item Health Belief Model-based scale included in the original Karki et al. instrument. This scale measures four domains: perceived benefits, perceived severity, perceived barriers, and perceived susceptibility. Responses were rated on a five-point Likert scale (1 = strongly disagree to 5 = strongly agree), with higher scores reflecting stronger beliefs in each respective domain.

HPV vaccine uptake was measured by asking, "Have you received the HPV vaccine?" Response options were "Yes," "No," "Maybe," and "Probably not." Unvaccinated participants were also asked about their willingness to receive the vaccine and their reasons for acceptance or refusal.

## Sampling and data collection procedures

A convenience sampling approach was adopted. Recruitment was conducted through official university-affiliated Telegram channels and student social media groups. Each of the five participating faculties disseminated the survey link via their formal communication platforms to reach students across all academic years. The survey instrument was hosted on Google Forms and could be completed at any time convenient to the participant.

Upon clicking the survey link, all eligible medical students were first presented with an informed consent page that described the study purpose, voluntary nature of participation, confidentiality assurances, and data usage. Participants could only proceed to the questionnaire after selecting the option "I agree to participate." Students who did not provide consent were automatically excluded. This procedure ensured documented electronic informed consent prior to data collection.

The sequence of participant recruitment, consent, and questionnaire completion is illustrated in Fig 1. All respondents first completed the socio-demographic section, followed by HPV awareness items. Students who reported prior awareness of HPV were presented with the HPV knowledge items, and subsequently the HPV vaccine awareness question. Participants aware of the HPV vaccine completed the HPV vaccine knowledge section, while all respondents, regardless of awareness level, answered items on vaccine acceptance and hesitancy. The flowchart also depicts the number of students excluded at each stage due to non-consent or lack of awareness of HPV or the HPV vaccine.

## Statistical analysis

All data were analyzed using IBM SPSS Statistics version 28 for Windows. Descriptive statistics (frequencies, percentages, means, standard deviations, medians, and interquartile ranges) were used to summarize participant characteristics, awareness, and knowledge levels. The Pearson chi-square test was applied to assess associations between categorical variables, while the Mann–Whitney U test was used for continuous variables with non-normal distribution, as determined by the Shapiro–Wilk test. Binary logistic regression was performed to identify independent predictors of HPV awareness, with variables entered using a forward stepwise likelihood ratio method. Adjusted odds ratios (AOR) and 95% confidence intervals (CI) were reported. Multicollinearity was assessed using the variance inflation factor (VIF), and model fit was evaluated using the Hosmer–Lemeshow goodness-of-fit test. Statistical significance was set at a two-tailed p-value of <0.05.

## Ethics approval and consent to participate

This study was conducted in accordance with the principles outlined in the Declaration of Helsinki. The study protocol was reviewed and approved by the Institutional Review Board (IRB) of the Faculty of Medicine, Mansoura University, under the approval number R.24.08.2763.R3. All participants were informed about the study objectives, procedures, potential benefits, and possible risks prior to participation. Informed electronic consent was obtained from all respondents through the survey platform before accessing the questionnaire. Because all participants were university students aged ≥18 years, parental or guardian consent was not required. Participation was entirely voluntary, and no monetary or non-monetary incentives were provided. Data were collected anonymously through a secure Google Forms platform, ensuring confidentiality and privacy throughout the research process.

## Results

### Participant characteristics and HPV awareness

The study cohort comprised 1,431 participants, stratified into those aware (n = 1,302, 91.0%) and unaware (n = 129, 9.0%) of HPV (Table 1). Comparative analyses demonstrated significant demographic and socioeconomic variations between groups. Participants with HPV awareness were older (median age 24 years, IQR 22–25 vs. 21 years, IQR 19–23;

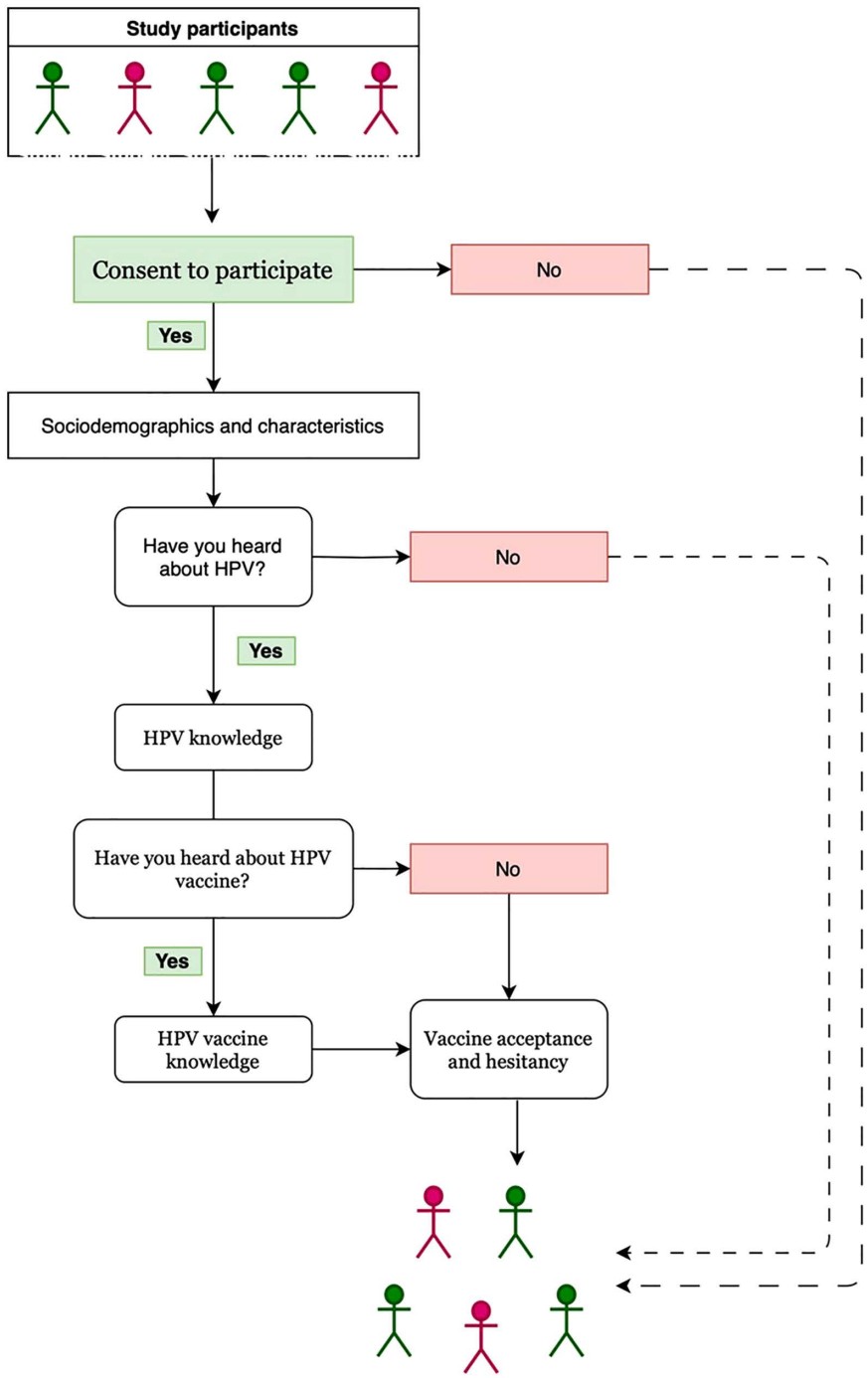

**Fig 1. Flowchart of participant recruitment and data collection process.** The diagram shows each stage of the study, including recruitment, consent, questionnaire completion, and exclusion criteria.

**Table 1. Sociodemographic and educational characteristics of study participants associated with HPV awareness.**

| Characteristics | HPV Awareness | | P-value[2] |
|---|---|---|---|
| | No N = 129[1] | Yes N = 1302[1] | |
| **Age** | 21 (19-23) | 24 (22-25) | **<0.001** |
| **Gender** | | | 0.13 |
| Female | 77 (60%) | 686 (53%) | |
| Male | 52 (40%) | 616 (47%) | |
| **Nationality** | | | **<0.001** |
| Egyptian | 104 (81%) | 1,189 (91%) | |
| Non-Egyptian | 25 (19%) | 113 (8.7%) | |
| **Residency** | | | 0.11 |
| Rural | 57 (44%) | 483 (37%) | |
| Urban | 72 (56%) | 819 (63%) | |
| **Marital Status** | | | 0.80 |
| Single | 123 (95%) | 1,219 (94%) | |
| Married | 6 (5%) | 78 (6.5%) | |
| Others | 0 (0%) | 5 (0.5%) | |
| **Year of study** | | | **<0.001** |
| First Year | 17 (13%) | 16 (1.2%) | |
| Second Year | 43 (33%) | 65 (5.0%) | |
| Third Year | 12 (9.3%) | 72 (5.5%) | |
| Fourth Year | 17 (13%) | 146 (11%) | |
| Fifth Year | 8 (6.2%) | 339 (26%) | |
| Internship Year | 32 (25%) | 664 (51%) | |
| **First-generation medical student** | 100 (78%) | 1,022 (78%) | 0.6 |
| **Highest level of father's education** | | | 0.14 |
| Less than a bachelor's degree | 39 (30%) | 324 (25%) | |
| Bachelor's degree (not related to medical field) | 56 (43%) | 705 (54%) | |
| Bachelor's degree (related to medical field) | 11 (8.5%) | 94 (7.2%) | |
| Postgraduate's degree | 23 (18%) | 179 (14%) | |
| **Highest level of mother's education** | | | **0.01** |
| Less than a bachelor's degree | 46 (36%) | 444 (34%) | |
| Bachelor's degree (not related to medical field) | 54 (42%) | 681 (52%) | |
| Bachelor's degree (related to medical field) | 17 (13%) | 88 (6.8%) | |
| Postgraduate's degree | 12 (9.3%) | 89 (6.8%) | |
| **Religious Affiliation** | | | – |
| Islam | 417 (100%) | 885 (100%) | |
| **How important is religion in your daily life?** | | | **0.01** |
| Not important | 3 (2.3%) | 9 (0.7%) | |
| Slightly important | 7 (5.4%) | 23 (1.8%) | |
| Moderately important | 14 (11%) | 174 (13%) | |
| Very important | 105 (81%) | 1,096 (84%) | |
| **Attended any lectures or seminars specifically about HPV** | 28 (22%) | 693 (53%) | **<0.001** |
| **Vaccination during infancy** | | | 0.06 |
| Don't know | 53 (41%) | 469 (36%) | |
| No | 15 (12%) | 263 (20%) | |
| Yes | 61 (47%) | 570 (44%) | |

*(Continued)*

**Table 1.** (Continued)

| Characteristics | HPV Awareness | | P-value[2] |
|---|---|---|---|
| | No N = 129[1] | Yes N = 1302[1] | |
| **Availability of local vaccination center** | | | 0.01 |
| Don't know | 69 (53%) | 565 (43%) | |
| No | 16 (12%) | 124 (9.5%) | |
| Yes | 44 (34%) | 613 (47%) | |
| **Family history of cervical cancer** | | | 0.001 |
| Don't know | 10 (7.8%) | 44 (3.4%) | |
| No | 109 (84%) | 1,209 (93%) | |
| Yes | 10 (7.8%) | 49 (3.8%) | |
| **Do you consider yourself as a religious person?** | | | 0.011 |
| Maybe | 51 (40%) | 504 (39%) | |
| No | 15 (12%) | 89 (6.8%) | |
| Yes | 63 (49%) | 709 (54%) | |

[1]Median (Q1, Q3); n (%), [2]Wilcoxon rank sum test; Pearson's Chi-squared test; Fisher's exact test. Values are presented as frequencies and column percentages. Percentages for religious affiliation total 100% within each awareness category, reflecting the homogeneity of the sample.

p < 0.001) and more likely to be Egyptian (91.3% vs. 81.0%; p < 0.001). Academic progression exhibited a strong positive association with awareness, with 51.0% of interns demonstrating awareness compared to 1.2% of first-year students (p < 0.001). Exposure to HPV-related educational sessions significantly predicted awareness (53.0% vs. 22.0%; p < 0.001), as did self-reported religious importance (84.0% vs. 81.0%; p = 0.01).

## Knowledge of HPV pathogenesis and clinical implications

General knowledge regarding HPV transmission was high, with 97–98% correctly identifying its sexual transmission route (Table 2). However, pervasive misconceptions were observed: 67% erroneously believed symptomatic presentation was typical (p = 0.8 for gender difference), while only 30–41% recognized the potential for spontaneous viral clearance (p < 0.001, female predominance). Oncogenic risk awareness was higher for cervical (95–97%; p = 0.06) than penile cancer (59–66%; p = 0.006), with males demonstrating superior knowledge of male-specific outcomes. Notably, 26% of females versus 20% of males incorrectly attributed cervical cancer etiology to genital warts (p = 0.014), and 50% versus 42% conflated HPV with herpesvirus pathogenesis (p = 0.004).

## Understanding of HPV vaccination parameters

Vaccine-related knowledge exhibited substantial gaps, particularly regarding indications and safety (Table 3). A majority (78%) incorrectly associated vaccination with serious adverse effects (p = 0.9), while 70–76% erroneously restricted eligibility to sexually active individuals (p = 0.05, male predominance). Most participants (83–85%) inappropriately concluded that vaccinated women could forego cervical cancer screening (p = 0.3). Although 85–88% correctly identified pre-sexual debut as the optimal vaccination timing (p = 0.2), only 53–54% recognized the multidose regimen requirement (p = 0.8).

## Multivariable predictors of HPV awareness

Logistic regression modeling revealed several independent determinants of awareness (Table 4). Non-Egyptian nationality conferred reduced likelihood (AOR 0.432, 95% CI 0.209–0.891; p = 0.023), whereas urban residency increased odds (AOR 1.792, 95% CI 1.101–2.917; p = 0.019). Academic seniority demonstrated a dose-response relationship, with interns

**Table 2. HPV knowledge items and correct response rates by Gender.**

| Items | Female N = 592[1] | Male N = 665[1] | P-value[2] |
|---|---|---|---|
| HPV is a sexually transmitted infection. | 579 (98%) | 646 (97%) | 0.5 |
| There is a cure for HPV. | 205 (35%) | 248 (37%) | 0.3 |
| Having one type of HPV means that you cannot acquire new types. | 541 (91%) | 625 (94%) | 0.07 |
| There is a screening that is commonly used to test males for HPV. | 271 (46%) | 313 (47%) | 0.6 |
| An abnormal Pap smear may indicate that a woman has HPV. | 491 (83%) | 560 (84%) | 0.5 |
| Most genital HPV infections do not clear up on their own. | 244 (41%) | 198 (30%) | **<0.001** |
| A person usually has symptoms when infected with HPV. | 394 (67%) | 446 (67%) | 0.8 |
| HPV is not a very common virus. | 448 (76%) | 511 (77%) | 0.6 |
| HPV infection can cause genital warts. | 567 (96%) | 637 (96%) | 0.9 |
| HPV infection can cause genital herpes. | 296 (50%) | 278 (42%) | **0.004** |
| Certain types of HPV can lead to cervical cancer in women. | 561 (95%) | 644 (97%) | 0.06 |
| HPV can lay dormant in the body for years without symptoms. | 511 (86%) | 575 (86%) | 0.9 |
| A person's chances of getting HPV increase with the number of sexual partners they have. | 566 (96%) | 628 (94%) | 0.3 |
| Most people with HPV have visible signs or symptoms of the infection. | 385 (65%) | 425 (64%) | 0.7 |
| Genital warts can cause cervical cancer. | 151 (26%) | 131 (20%) | **0.014** |
| Condoms are not effective in preventing HPV. | 441 (74%) | 464 (70%) | 0.063 |
| HPV can cause penile cancer. | 348 (59%) | 441 (66%) | **0.006** |
| HPV can cause anal cancer. | 358 (60%) | 412 (62%) | 0.6 |
| HPV can cause oropharyngeal cancer. | 373 (63%) | 453 (68%) | **0.05** |
| Nearly all sexually active men and women will contract HPV at some point. | 323 (55%) | 371 (56%) | 0.7 |

The "Items" column lists the survey questions only; correct answers are not indicated. Values represent the number and percentage of respondents who selected the correct answer for each item. [1] n: correct answer (%), [2] Pearson's Chi-squared test.

showing 15-fold greater awareness than first-year students (p<0.001). Didactic exposure to HPV content tripled awareness probability (AOR 3.349, 95% CI 1.982–5.659; p<0.001), while religious devotion exhibited a modest association (AOR 1.82, 95% CI 1.06–3.12; p=0.03).

## Gender-specific attitudes toward vaccination

Significant gender disparities emerged in vaccination receptivity and risk perception (Table 5). Males exhibited greater willingness to vaccinate (44% vs. 36%; p=0.003) despite lower perceived personal risk (25% vs. 33% strongly disagreeing about infection risk; p<0.001). Females more frequently endorsed safety concerns (7.6% vs. 3.9% considering vaccines unsafe; p=0.005) and vaccination-related stigma (18% vs. 12% reporting embarrassment; p=0.019). Paradoxically, females demonstrated heightened cancer threat awareness, with 23% versus 17% strongly agreeing about HPV-associated cancer lethality (p=0.003).

## Comparison by educational level

Additional analyses were performed to compare HPV knowledge, attitudes, and practices between preclinical and clinical medical students (S1 Table, S2 Table, and S3 Table). Clinical students generally exhibited higher knowledge scores across most HPV-related items. They were significantly more likely to recognize that HPV is a sexually transmitted infection (98% vs. 95%, p=0.058), that certain HPV types cause cervical cancer (96% vs. 92%, p=0.013), and that HPV can lead to penile (p=0.012) and anal (p=0.015) cancers. However, misconceptions persisted; more than half of both groups incorrectly believed that HPV infection has visible symptoms.

**Table 3. HPV vaccine knowledge and correct response rates by Gender.**

| Items | Female N = 387[1] | Male N = 463[1] | P-value[2] |
|---|---|---|---|
| The HPV vaccination is recommended only for women and girls. | 251 (65%) | 319 (69%) | 0.2 |
| The HPV vaccination is often associated with serious side-effects. | 302 (78%) | 359 (78%) | 0.9 |
| The HPV vaccine is given as a single shot. | 206 (53%) | 250 (54%) | 0.8 |
| It is best to receive the HPV shot before being sexually active. | 329 (85%) | 409 (88%) | 0.2 |
| It is too late for teenagers who already had sex to have the HPV vaccine. | 308 (80%) | 389 (84%) | 0.094 |
| If a woman has obtained HPV vaccination, she will not need to do the pap smears. | 320 (83%) | 394 (85%) | 0.3 |
| The HPV vaccine will prevent all causes of HPV associated cancers. | 243 (63%) | 314 (68%) | 0.12 |
| HPV vaccine is offered only to sexually active people. | 272 (70%) | 352 (76%) | **0.05** |

[1]n: correct answer (%), [2]Pearson's Chi-squared test.

**Table 4. Multivariate logistic regression analysis of factors associated with HPV awareness.**

| Variable | AOR | 95% CI | | P-value |
|---|---|---|---|---|
| | | Lower | Upper | |
| Nationality (Non-Egyptian vs Egyptian) | 0.432 | 0.209 | 0.891 | **0.023** |
| Residency (Urban vs Rural) | 1.792 | 1.101 | 2.917 | **0.019** |
| Year of Study (First Year vs Reference) | 0.067 | 0.016 | 0.274 | **<0.001** |
| Year of Study (Fourth Year vs Reference) | 0.269 | 0.103 | 0.703 | **0.008** |
| Year of Study (Second Year vs Reference) | 0.078 | 0.025 | 0.244 | **<0.001** |
| Year of Study (Third Year vs Reference) | 0.283 | 0.088 | 0.913 | **0.03** |
| Religion Importance (Not Important vs Reference) | 0.141 | 0.024 | 0.821 | **0.03** |
| Religion Importance (Slightly Important vs Reference) | 0.134 | 0.036 | 0.504 | **0.003** |
| Attended HPV Lectures/Seminars (Yes vs No) | 3.349 | 1.982 | 5.659 | **<0.001** |

Preclinical students showed lower awareness regarding the absence of a screening test for men ($p = 0.005$) and were more likely to believe that condoms are ineffective in preventing HPV transmission ($p < 0.001$). They also expressed more uncertainty about whether HPV infections have a cure ($p = 0.026$). Attitudinal differences were also evident. Preclinical students were more likely to believe the HPV vaccine has serious side effects ($p < 0.001$), that vaccinated women do not need Pap smears ($p = 0.030$), and that it is too late for sexually active teenagers to receive the vaccine ($p = 0.020$). They were also more likely to feel embarrassed about receiving an HPV vaccine because it is for an STI ($p = 0.004$).

Regarding awareness and practices, 70% of clinical students had heard about the HPV vaccine compared with 53% of preclinical students ($p < 0.001*$). Overall vaccination experience did not differ significantly between groups ($p = 0.40*$). These results indicate that clinical students have a broader understanding of HPV and its prevention, yet misconceptions and vaccine hesitancy remain common in both educational stages, underscoring the need for structured educational interventions throughout medical training.

## Discussion

This national cross-sectional study aimed to provide a comprehensive assessment of awareness, knowledge, and beliefs regarding HPV and its vaccine among Egyptian medical students. By sampling students from different academic years and regions, the study offers a detailed picture of how demographic, educational, and cultural factors shape HPV-related understanding and attitudes in future healthcare providers. The findings reveal a high level of self-reported awareness, but also substantial misconceptions about HPV pathogenesis, its oncogenic potential, and the safety and eligibility criteria of

**Table 5. Attitudes, beliefs and perceptions toward HPV and HPV vaccine by Gender.**

| Variable | Female N = 592[1] | Male N = 665[1] | P-value[2] |
|---|---|---|---|
| **The HPV vaccine will protect from getting HPV associated cancers.** | | | 0.5 |
| Agree | 224 (38%) | 249 (37%) | |
| Disagree | 38 (6.4%) | 40 (6.0%) | |
| Not Sure | 185 (31%) | 214 (32%) | |
| Strongly Agree | 79 (13%) | 72 (11%) | |
| Strongly Disagree | 66 (11%) | 90 (14%) | |
| **The HPV vaccine will be effective in preventing HPV infection.** | | | 0.13 |
| Agree | 247 (42%) | 271 (41%) | |
| Disagree | 61 (10%) | 60 (9.0%) | |
| Not Sure | 135 (23%) | 162 (24%) | |
| Strongly Agree | 94 (16%) | 85 (13%) | |
| Strongly Disagree | 55 (9.3%) | 87 (13%) | |
| **Getting an HPV vaccine will benefit my health.** | | | 0.14 |
| Agree | 231 (39%) | 249 (37%) | |
| Disagree | 45 (7.6%) | 52 (7.8%) | |
| Not Sure | 146 (25%) | 153 (23%) | |
| Strongly Agree | 108 (18%) | 109 (16%) | |
| Strongly Disagree | 62 (10%) | 102 (15%) | |
| **If I have an HPV infection, it would be disruptive to my health.** | | | 0.3 |
| Agree | 195 (33%) | 212 (32%) | |
| Disagree | 99 (17%) | 106 (16%) | |
| Not Sure | 169 (29%) | 188 (28%) | |
| Strongly Agree | 73 (12%) | 70 (11%) | |
| Strongly Disagree | 56 (9.5%) | 89 (13%) | |
| **If I have HPV-associated cancer, it would threaten the relationship with my boyfriend, husband, or partner.** | | | **0.008** |
| Agree | 183 (31%) | 224 (34%) | |
| Disagree | 51 (8.6%) | 58 (8.7%) | |
| Not Sure | 146 (25%) | 150 (23%) | |
| Strongly Agree | 149 (25%) | 126 (19%) | |
| Strongly Disagree | 63 (11%) | 107 (16%) | |
| **HPV-associated cancer is a life threating disease.** | | | **0.003** |
| Agree | 194 (33%) | 210 (32%) | |
| Disagree | 74 (13%) | 68 (10%) | |
| Not Sure | 127 (21%) | 174 (26%) | |
| Strongly Agree | 135 (23%) | 111 (17%) | |
| Strongly Disagree | 62 (10%) | 102 (15%) | |
| **I think the HPV vaccine is unsafe.** | | | **0.005** |
| Agree | 45 (7.6%) | 26 (3.9%) | |
| Disagree | 214 (36%) | 294 (44%) | |
| Not Sure | 217 (37%) | 215 (32%) | |
| Strongly Agree | 9 (1.5%) | 7 (1.1%) | |
| Strongly Disagree | 107 (18%) | 123 (18%) | |

*(Continued)*

**Table 5.** (Continued)

| Variable | Female N = 592[1] | Male N = 665[1] | P-value[2] |
|---|---|---|---|
| **I feel embarrassed to get an HPV vaccine because it is for a sexually transmitted infection.** | | | **0.019** |
| Agree | 104 (18%) | 78 (12%) | |
| Disagree | 165 (28%) | 209 (31%) | |
| Not Sure | 158 (27%) | 196 (29%) | |
| Strongly Agree | 27 (4.6%) | 20 (3.0%) | |
| Strongly Disagree | 138 (23%) | 162 (24%) | |
| **It is hard to find a provider or clinic that has the vaccine available.** | | | 0.2 |
| Agree | 133 (22%) | 128 (19%) | |
| Disagree | 93 (16%) | 117 (18%) | |
| Not Sure | 285 (48%) | 304 (46%) | |
| Strongly Agree | 31 (5.2%) | 37 (5.6%) | |
| Strongly Disagree | 50 (8.4%) | 79 (12%) | |
| **I am concerned that the HPV vaccine costs more than my parents or I can pay.** | | | 0.6 |
| Agree | 88 (15%) | 91 (14%) | |
| Disagree | 132 (22%) | 160 (24%) | |
| Not Sure | 294 (50%) | 315 (47%) | |
| Strongly Agree | 13 (2.2%) | 23 (3.5%) | |
| Strongly Disagree | 65 (11%) | 76 (11%) | |
| **I am at risk of contracting HPV.** | | | **<0.001** |
| Agree | 57 (9.6%) | 43 (6.5%) | |
| Disagree | 173 (29%) | 262 (39%) | |
| Not Sure | 158 (27%) | 184 (28%) | |
| Strongly Agree | 10 (1.7%) | 9 (1.4%) | |
| Strongly Disagree | 194 (33%) | 167 (25%) | |
| **I am at risk for getting cervical cancer.** | | | **<0.001** |
| Agree | 46 (7.8%) | 41 (6.2%) | |
| Disagree | 124 (21%) | 251 (38%) | |
| Not Sure | 109 (18%) | 221 (33%) | |
| Strongly Agree | 10 (1.7%) | 10 (1.5%) | |
| Strongly Disagree | 303 (51%) | 142 (21%) | |
| **Have you received any type of vaccination?** | 475 (80%) | 562 (85%) | **0.046** |
| **Are you willing to take the HPV vaccine?** | 213 (36%) | 294 (44%) | **0.003** |

[1]n (%), [2]Pearson's Chi-squared test.

the vaccine. These gaps highlight the need for targeted educational strategies that address both factual knowledge and sociocultural perceptions.

A recently published study by Abdelaziz et al. (2025) evaluated Egyptian medical students' knowledge and attitudes regarding the role of the HPV vaccine in preventing oropharyngeal cancer [17]. While both studies address the topic of HPV awareness among medical students, the present work differs in several respects. Our study assesses broader domains of HPV-related knowledge, beliefs, and attitudes toward vaccination, covering cervical, oropharyngeal, and other HPV-associated cancers, and explores sociodemographic and behavioral predictors influencing awareness and vaccine acceptance. In addition, our sample included participants from multiple Egyptian universities and academic years, providing a wider representation of medical students. Therefore, this study offers

complementary evidence that expands upon previous findings and highlights persistent gaps in HPV education and vaccination awareness in Egypt.

The high prevalence of HPV awareness (91%) in this cohort is encouraging and aligns with reports from other medical student populations in the Middle East and North Africa (MENA) region [20,21]. However, the association between awareness and academic seniority suggests that much of this knowledge is acquired during later clinical years [22]. Interns demonstrated markedly higher awareness than first-year students, indicating that formal HPV-related education is likely concentrated in advanced stages of training [23]. Similar trends have been documented in Jordan, Saudi Arabia, and other countries where sexual and reproductive health topics are introduced late in the curriculum [20,21]. Integrating HPV content earlier could ensure that junior students enter clinical environments with accurate information, enabling them to engage effectively in prevention advocacy from the outset [24].

Despite strong recognition of HPV's sexual transmission route, important misconceptions persist. Two-thirds of participants believed the infection typically presents with symptoms, reflecting limited understanding of its often silent nature. This misconception, observed in other contexts such as Morocco [25], may lead students to underestimate the infection's public health significance and transmissibility. Similarly, the low proportion recognizing the possibility of spontaneous viral clearance indicates insufficient familiarity with HPV's natural history [26]. Addressing these misunderstandings is essential, as they influence both patient counseling and screening recommendations.

The finding that nearly half of students confused HPV with herpesvirus is concerning, as it suggests a gap in basic virology and sexually transmitted infection (STI) differentiation [27]. Overlapping modes of transmission may contribute to this conflation, but the persistence of such a fundamental error in a medical student cohort underscores the need for clearer, clinically integrated teaching about STIs [27]. Without such clarification, misinformation could be inadvertently perpetuated to patients.

Students' tendency to associate HPV primarily with cervical cancer, while scientifically accurate, reflects a limited and gendered framing of the virus. Although over 95% made this connection, fewer recognized HPV's role in male-specific cancers such as penile and anal cancer or in oropharyngeal cancer (OPC). This pattern is consistent with global public health messaging, which has historically focused on HPV as a women's health issue [28]. However, rising rates of HPV-related OPC, particularly among men, make it imperative that both genders are included in prevention narratives [17,20].

Vaccine-related knowledge revealed notable deficiencies. Many participants incorrectly believed that the HPV vaccine carries serious adverse effects or is restricted to individuals already sexually active. These misconceptions mirror attitudes reported in other MENA countries, where hesitancy has been linked to safety concerns, cultural beliefs, limited public awareness, and inadequate physician recommendation [29,30]. Additionally, the finding that most students believed vaccinated women could forego cervical cancer screening suggests gaps in understanding the complementary roles of vaccination and screening in prevention strategies [31].

Gender-based differences in vaccine attitudes point to the complex interplay between knowledge, perceived risk, and cultural context. Male students expressed greater willingness to vaccinate but perceived themselves at lower personal risk, while female students were more likely to acknowledge HPV's cancer threat yet reported higher safety concerns and stigma. This paradox has been noted in other cultural settings where norms around modesty, sexual behavior, and reputation influence women's health decisions even when knowledge levels are high [32,33]. Such findings underscore the importance of culturally sensitive educational interventions that address not only scientific facts but also the social narratives influencing decision-making.

Multivariable analysis identified urban residence, higher academic year, and prior HPV-related education as strong predictors of awareness. These associations point to practical intervention opportunities. Early integration of HPV education into the medical curriculum, equitable access to information for students in rural settings, and structured educational campaigns to dispel myths could collectively improve both knowledge and vaccine acceptance.

## Limitations

This study has several limitations. First, its cross-sectional design captures associations at a single point in time and therefore cannot determine whether increased educational exposure actually leads to greater HPV awareness or more favorable attitudes. While differences across academic years were observed, these reflect correlations rather than causal effects, as longitudinal data would be required to confirm directional changes over time. Second, reliance on self-reported data introduces potential recall and social desirability bias, particularly given the cultural sensitivity of sexual health topics in Egypt. Third, the online survey format may have excluded students with limited internet access, potentially underrepresenting rural populations. Fourth, although the questionnaire was adapted from a validated tool, certain misconceptions may have been under- or overestimated due to question interpretation. Finally, while the study included participants from multiple medical faculties nationwide, findings may not be generalizable to non-medical student populations or other healthcare disciplines.

## Conclusion

This national study reveals that while HPV awareness among Egyptian medical students is high, substantial gaps remain in understanding its natural history, broader cancer associations, and vaccine safety and eligibility. Awareness was strongly associated with academic seniority, urban residence, and prior education, indicating the need for earlier and more equitable curriculum integration. Misconceptions—such as symptom-based recognition, conflation with herpesvirus, and undervaluing male cancer risks—underscore the importance of comprehensive, evidence-based instruction. Gender differences in vaccine attitudes highlight the influence of sociocultural factors on health decision-making, even in medically trained populations. Strengthening HPV education from the early years of medical training and addressing cultural barriers can enhance vaccine acceptance and prepare future physicians to lead effective HPV prevention efforts.

## Supporting information

**S1 Table.  Comparison of HPV knowledge between preclinical and clinical medical students.**
(PDF)

**S2 Table.  Comparison of attitudes toward HPV and the HPV vaccine between preclinical and clinical medical students.**
(PDF)

**S3 Table.  Comparison of HPV-related awareness and vaccination practices between preclinical and clinical medical students.**
(PDF)

## Author contributions

**Conceptualization:** Muhammad Elmanzlawey, Abdel-Hady El-Gilany.

**Data curation:** Muhammad Elmanzlawey, Mohamed Terra, Mohamed Baklola, Abdel-Hady El-Gilany.

**Formal analysis:** Muhammad Elmanzlawey, Mohamed Terra, Mohamed Baklola.

**Resources:** Yara Ahmed, Ibrahim Noureddin Al-Kurd.

**Software:** Mohamed Terra.

**Supervision:** Mohamed Terra, Abdel-Hady El-Gilany.

**Validation:** Mohamed Baklola, Yara Ahmed, Mariam Abuelela, Ibrahim Noureddin Al-Kurd.

**Visualization:** Mohamed Baklola, Mariam Abuelela, Ibrahim Noureddin Al-Kurd.

**Writing – original draft:** Muhammad Elmanzlawey, Yara Ahmed, Mariam Abuelela, Ibrahim Noureddin Al-Kurd.

**Writing – review & editing:** Muhammad Elmanzlawey, Mohamed Baklola, Yara Ahmed, Mariam Abuelela, Ibrahim Noureddin Al-Kurd, Abdel-Hady El-Gilany.

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
