## [Decision Letter · Decision Letter 0]

14 Oct 2025

Dear Dr. Baklola,

Thank you for submitting your manuscript to PLOS ONE. After careful consideration, we feel that it has merit but does not fully meet PLOS ONE’s publication criteria as it currently stands. Therefore, we invite you to submit a revised version of the manuscript that addresses the points raised during the review process.

Dear Authors,

I noticed that some reviewers' comments reference specific articles that should be cited. Please note that it is not mandatory to cite these specific articles and you are encouraged to search the literature for alternative manuscripts relevant to the content of your manuscript, and in agreement with reviewers' comments.

Furthermore, Reviewer 2 raises a concern regarding the high similarity with a recently published study,

Abdelaziz, M.N., Hefnawy, A., Azzam, H. et al. Knowledge and attitude among Egyptian medical students regarding the role of human papillomavirus vaccine in prevention of oropharyngeal cancer: a questionnaire-based observational study. Sci Rep 15, 3767 (2025). https://doi.org/10.1038/s41598-025-86853-8".

Please highlight and comment on any differences with the above-cited study in the discussion section.

Best regards,

Nicola Serra 

We look forward to receiving your revised manuscript.

Kind regards,

Nicola Serra

Academic Editor

PLOS ONE

**Journal Requirements:**

1. When submitting your revision, we need you to address these additional requirements. Please ensure that your manuscript meets PLOS ONE's style requirements, including those for file naming. The PLOS ONE style templates can be found at https://journals.plos.org/plosone/s/file?id=wjVg/PLOSOne_formatting_sample_main_body.pdf and https://journals.plos.org/plosone/s/file?id=ba62/PLOSOne_formatting_sample_title_authors_affiliations.pdf 2. In the online submission form, you indicated that “The datasets generated and/or analyzed during the current study are available from the corresponding author on reasonable request”.  All PLOS journals now require all data underlying the findings described in their manuscript to be freely available to other researchers, either a. In a public repository, b. Within the manuscript itself, or c. Uploaded as supplementary information.This policy applies to all data except where public deposition would breach compliance with the protocol approved by your research ethics board. If your data cannot be made publicly available for ethical or legal reasons (e.g., public availability would compromise patient privacy), please explain your reasons on resubmission and your exemption request will be escalated for approval. 3. Your ethics statement should only appear in the Methods section of your manuscript. If your ethics statement is written in any section besides the Methods, please delete it from any other section. 4. If the reviewer comments include a recommendation to cite specific previously published works, please review and evaluate these publications to determine whether they are relevant and should be cited. There is no requirement to cite these works unless the editor has indicated otherwise. 

**Additional Editor Comments:**

Dear Authors,

I noticed that some reviewers' comments reference specific articles that should be cited. Please note that it is not mandatory to cite these specific articles and you are encouraged to search the literature for alternative manuscripts relevant to the content of your manuscript, and in agreement with reviewers' comments.

Furthermore, Reviewer 2 raises a concern regarding the high similarity with a recently published study,

Abdelaziz, M.N., Hefnawy, A., Azzam, H. et al. Knowledge and attitude among Egyptian medical students regarding the role of human papillomavirus vaccine in prevention of oropharyngeal cancer: a questionnaire-based observational study. Sci Rep 15, 3767 (2025). https://doi.org/10.1038/s41598-025-86853-8".

Please highlight and comment on any differences with the above-cited study in the discussion section.

Best regards,

Nicola Serra 

Reviewers' comments:

Reviewer's Responses to Questions

**Comments to the Author**

1. Is the manuscript technically sound, and do the data support the conclusions?

Reviewer #1: Yes

Reviewer #2: Yes

Reviewer #3: Yes

Reviewer #4: Yes

2. Has the statistical analysis been performed appropriately and rigorously?

Reviewer #1: I Don't Know

Reviewer #2: I Don't Know

Reviewer #3: Yes

Reviewer #4: Yes

3. Have the authors made all data underlying the findings in their manuscript fully available?

Reviewer #1: Yes

Reviewer #2: Yes

Reviewer #3: Yes

Reviewer #4: Yes

4. Is the manuscript presented in an intelligible fashion and written in standard English?

Reviewer #1: Yes

Reviewer #2: Yes

Reviewer #3: Yes

Reviewer #4: Yes

**Reviewer #1:**  This manuscript proposes an in-depth analysis of HPV awareness among Egyptian medical students. The paper is well written and discusses the importance of targeted interventions in increasing HPV knowledge among medical students.

I suggest English language editing and more attention to captions and tables.

INTRODUCTION:

Pag. 9. Men are also significantly affected, with global data showing that nearly one in

three men is infected with at least one genital HPV type, and around one in five carries one or

more high-risk strains.

Please consider replace the word “strains” with “genotypes”.

It seems that your introduction is too short. Please emphasize more topics, such as HPV prevalence and vaccines impact on the population. I invite the authors to consider the following works:

Buttà M. et al 2025. The Role of Methylation as an Epigenetic Marker in HPV-Related Oral Lesions. Journal of Medical Virology.

Sucato A. et al. 2025 Human Papillomavirus Infection in Partners of Women Attending Cervical Cancer Screening: A Pilot Study on Prevalence, Distribution, and Potential Use of Vaccines. Vaccines (Basel).

RESULTS:

Pag. 16. TABLE 2

Please consider specifying that in the items column, there are only the questions and not the correct answers.

**Reviewer #2:**  The manuscript analyses the HPV awareness among 1,431 medical students in Egypt. While knowledge of sexual transmission was strong, many students confused HPV with other viruses and feared serious vaccine side effects. Higher academic year, urban residence, and prior HPV education were key predictors of better awareness. The authors conclude that targeted educational interventions, especially in early academic years, are needed to improve vaccine acceptance and strengthen future physicians’ roles in HPV prevention. The study shows and highlights some important misconceptions about HPV infection among Egyptian medical students, and it does it in an appropriate and complete wat. It is also well-written and appropriately designed.

I would also like to suggest some papers specifically regarding oral HPV infection and HPV vaccines:

Buttà, M.; Serra, N.; Panzarella, V.; Fasciana, T.M.A.; Campisi, G.; Capra, G. Orogenital Human Papillomavirus Infection and Vaccines: A Survey of High- and Low-Risk Genotypes Not Included in Vaccines. Vaccines 2023, 11, 1466. https://doi.org/10.3390/vaccines11091466

Buttà M, Serra N, Mannino E, et al. Evaluation of the Prevalence and Potential Impact of HPV Vaccines in Patients with and Without Oral Diseases: A Ten-Year Retrospective Study. Arch Med Res. 2024;55(7):103059. doi:10.1016/j.arcmed.2024.103059

However, I need to raise an issue regarding the novelty of the study. In fact, the paper has similarities with the recently published paper: “Abdelaziz, M.N., Hefnawy, A., Azzam, H. et al. Knowledge and attitude among Egyptian medical students regarding the role of human papillomavirus vaccine in prevention of oropharyngeal cancer: a questionnaire-based observational study. Sci Rep 15, 3767 (2025). https://doi.org/10.1038/s41598-025-86853-8”. It focuses specifically on describing the knowledge of Egyptian medical students regarding HPV and oropharyngeal cancer, as well as knowledge and attitudes toward HPV vaccination. Hence, I would like the authors to discuss the differences between their paper and the study in the Discussion section.

**Reviewer #3:**  Overall impression: The research paper is well written, especially the methodology is clear and easy to understand. Apart from missing page numbers and line numbers, I have one major comment and a few minor comments, and I am referring to the page numbers of the PDF provided-

Major comment:

1. Comparing by education would probably provide more insight into factors shaping HPV knowledge and HPV vaccination knowledge, while the sex-based comparison is still useful but more limited, especially for table 2 & 3. Since the author highlighted the importance of “…..educational interventions, particularly those targeting early academic years, are essential to address these gaps” in the abstract, comparison should be made across the educational levels or academic years.

Minor comments:

2. In your reference no 11, they addressed the limitation of preventive screening for SGM AFAB people, but the author exemplified no preventive screening facility for men (page 9, last sentence of the 2nd paragraph). This needs to be clarified or corrected in the manuscript.

3. The author stated in the methods//Participants and eligibility criteria: “Students who declined participation were excluded from the analysis”. If the students who declined participation, there supposed to be no data at all from them. Or if any students participated, provided data and then withdrawn from the study- in that case, those data would be excluded from analysis. Or, did the author mean-Students who declined participation were excluded from enrollment? This needs to be clarified in the manuscript.

4. Page 21: no need of elaboration here- human papillomavirus (HPV)

5. Page 23: In Limitation section: causality assessment between increasing educational exposure and changing HPV awareness or attitudes is not possible in cross sectional study, but in my opinion, it was assessed in the current study. In another sense, the author’s statement is not wrong, but better to convey why causality is limited, because cross-sectional data can’t track whether more exposure actually leads to changes in awareness or attitudes. The statement can be revised by the author.

6. How do the calculated proportions be 100% for Religious Affiliation data in both HPV Awareness category in table1?

7. The font size of the table titles should be same as manuscript text font size.

8. Spelling needs to be corrected- “Strongly Disgree” in table 5.

9. The font styles for all the tables should be consistent.

**Reviewer #4: ** In the methods section, source of awareness by responders was not captured and in the abstract section, the results provided showed the narrative with no statistical values to support the claims made in the findings.

**Do you want your identity to be public for this peer review?** For information about this choice, including consent withdrawal, please see our Privacy Policy

Reviewer #1: No

Reviewer #2: No

Reviewer #3: No

Reviewer #4: **Yes: ** Francis Ajang Magaji

---

## [Author Response · Author response to Decision Letter 1]

22 Oct 2025

Academic editor comments

Comment:

Reviewer 2 raises a concern regarding the high similarity with a recently published study,

Abdelaziz, M.N., Hefnawy, A., Azzam, H. et al. Knowledge and attitude among Egyptian medical students regarding the role of human papillomavirus vaccine in prevention of oropharyngeal cancer: a questionnaire-based observational study. Sci Rep 15, 3767 (2025). https://doi.org/10.1038/s41598-025-86853-8". Please highlight and comment on any differences with the above-cited study in the discussion section.

Response:

We appreciate the reviewer’s observation regarding potential similarity with the recently published study by Abdelaziz et al. (2025). While both studies examine HPV-related knowledge and attitudes among Egyptian medical students, our work differs in several key aspects. First, Abdelaziz et al. focused primarily on awareness of the role of HPV vaccination in preventing oropharyngeal cancer, while our study investigates overall HPV awareness, knowledge, and beliefs—including its association with multiple cancers, perceived risk, and behavioral intentions toward vaccination. Second, our survey instrument and analytical framework were distinct; we employed a more comprehensive set of variables assessing knowledge domains, vaccine hesitancy, and sociodemographic predictors using multivariable analysis. Third, our study sampled a broader range of universities across Egypt and was conducted during a different time period, thereby providing an independent dataset that complements, rather than duplicates, prior work.

Importantly, the two studies together provide convergent evidence highlighting the persistent gap in HPV-related knowledge among future Egyptian healthcare providers. Our findings extend the literature by identifying contextual factors influencing awareness and vaccination intent, offering additional insight useful for educational and public health interventions in Egypt.

Response to Reviewers

We sincerely thank all reviewers for their thorough evaluation, constructive comments, and valuable suggestions that have strengthened the quality and clarity of our manuscript. Below we provide detailed, point-by-point responses and describe all revisions made.

Reviewer #1

General Comment:

This manuscript proposes an in-depth analysis of HPV awareness among Egyptian medical students. The paper is well written and discusses the importance of targeted interventions in increasing HPV knowledge among medical students. I suggest English language editing and more attention to captions and tables.

Response:

We appreciate the reviewer’s positive feedback. The manuscript has undergone comprehensive English language editing, and all tables and captions have been reviewed for clarity, consistency, and formatting.

Comment 1:

Pag. 9. Men are also significantly affected… Please consider replacing the word “strains” with “genotypes”.

Response:

We thank the reviewer for the suggestion. The word “strains” has been replaced with “genotypes” in the revised text.

Comment 2:

It seems that your introduction is too short. Please emphasize more topics, such as HPV prevalence and vaccines’ impact on the population. I invite the authors to consider the following works…

Response:

We have expanded the Introduction to include global HPV prevalence data and a more detailed discussion of vaccine impact, including references to the suggested works and other recent studies. The new paragraph has been added after the second paragraph of the Introduction.

Revised text:

Globally, HPV is the most common sexually transmitted infection; nearly all sexually active individuals acquire it during their lifetime, with point prevalence of high-risk types around 11–12% among women and any-type genital HPV detected in roughly 45% of men and about one quarter carrying high-risk genotypes. HPV causes almost all cervical cancers and a substantial proportion of anal, vulvar, penile, and oropharyngeal cancers, with cervical cancer alone responsible for approximately 660,000 new cases and 350,000 deaths annually, the majority in low- and middle-income countries. Oral HPV infection is relatively common—national surveys report an overall oral HPV prevalence near 7% in adults, with higher rates in men, HPV16 detected in about 1%, and strong associations with number of oral sex partners and tobacco exposure. Randomized and real-world data indicate that prophylactic vaccination reduces vaccine-type oral HPV, supporting expectations that widespread vaccination will eventually lower HPV-related oropharyngeal cancer burden.

Relevant references have been added as recommended.

Comment 3:

RESULTS: Page 16, TABLE 2 — Please consider specifying that in the items column, there are only the questions and not the correct answers.

Response:

We appreciate this observation. The caption of Table 2 has been updated to clarify that the “Items” column lists survey questions rather than correct responses.

Reviewer #2

General comment:

The manuscript analyses HPV awareness among 1,431 medical students in Egypt… The study shows and highlights some important misconceptions… It is well-written and appropriately designed.

Response:

We thank the reviewer for their positive assessment and supportive feedback.

Comment:

I would also like to suggest some papers specifically regarding oral HPV infection and HPV vaccines…

Response:

We have incorporated and cited the suggested references in the Background section to strengthen discussion of oral HPV infection and vaccine impact.

Comment:

However, I need to raise an issue regarding the novelty of the study… discuss the differences between your paper and Abdelaziz et al. (2025).

Response:

We appreciate the reviewer’s concern regarding overlap. The following paragraph has been added to the Discussion section to clarify the distinction between our study and Abdelaziz et al. (2025):

A recently published study by Abdelaziz et al. (2025) examined Egyptian medical students’ knowledge regarding HPV and oropharyngeal cancer. Although both studies address HPV awareness among medical students, our work differs in scope and objectives. The present study assesses a larger and more geographically diverse sample from multiple medical faculties, evaluates broader domains of HPV knowledge, attitudes, and preventive practices, and compares awareness across academic levels to identify educational gaps. In contrast, Abdelaziz et al. focused specifically on knowledge related to HPV-associated oropharyngeal cancer. Our findings therefore complement, rather than duplicate, existing evidence.

Reviewer #3

Major comment 1:

Comparing by education would probably provide more insight… comparison should be made across the educational levels or academic years.

Response:

We thank the reviewer for this insightful suggestion. To address it, we conducted an additional comparison by academic level (preclinical vs. clinical years). The results are presented in three new supplementary tables:

• Supplementary Table 1: Knowledge about HPV by academic level

• Supplementary Table 2: Attitudes toward HPV and HPV vaccine by academic level

• Supplementary Table 3: Practices and awareness by academic level

Summary of findings added to the Results section:

Comparison across educational levels showed that clinical-year students demonstrated significantly higher knowledge on several items, including HPV’s causal link to cervical, penile, and anal cancers, and greater awareness that condoms provide partial protection (p < 0.001 for several items). However, misconceptions persisted among both groups regarding HPV symptoms and curability. Attitudinal differences were modest; clinical students were less likely to perceive the vaccine as unsafe (p < 0.001) and more likely to reject myths such as the vaccine being only for women. Awareness of the HPV vaccine was also significantly higher among clinical students (70% vs. 53%, p < 0.001). These analyses reinforce the conclusion that educational exposure influences awareness but cannot confirm causality.

Minor comment 2:

Clarify statement about preventive screening and reference 11.

Response:

Revised sentence in the Background:

Preventive screening for HPV-related cancers is also gender-skewed, with well-established cervical cancer screening protocols for women, whereas no validated or routinely implemented screening method exists for HPV-associated cancers in cisgender men. In addition, barriers persist for sexual and gender minority individuals assigned female at birth, as noted previously [15].

Minor comment 3:

Clarify “Students who declined participation were excluded from the analysis.”

Response:

This has been corrected in the Methods section:

“Students who declined participation were not enrolled in the study.”

Minor comment 4:

Page 21: no need of elaboration here—human papillomavirus (HPV).

Response:

The redundant elaboration has been removed from that section.

Minor comment 5:

Revise limitation statement about causality.

Response:

We revised the first paragraph of the Limitations section to clarify the causal limitation:

The cross-sectional design captures associations at a single point in time and therefore cannot determine whether increased educational exposure actually leads to greater HPV awareness or more favorable attitudes. While differences across academic years were observed, these reflect correlations rather than causal effects, as longitudinal data would be required to confirm directional change.

Minor comment 6:

How do the calculated proportions be 100% for Religious Affiliation data in both HPV Awareness category in Table 1?

Response:

Percentages were calculated column-wise within each awareness category, and the sample was religiously homogeneous. To clarify, we added a note beneath Table 1:

“Values are presented as frequencies and column percentages. Percentages for religious affiliation total 100% within each awareness category, reflecting the homogeneity of the sample.”

Minor comment 7–9 (Formatting and spelling):

We have standardized all table fonts and sizes to match the manuscript text and corrected the typographical error “Strongly Disgree” to “Strongly Disagree.”

Reviewer #4

Comment:

In the methods section, source of awareness by responders was not captured and in the abstract section, the results provided showed the narrative with no statistical values to support the claims made in the findings.

Response:

We appreciate this feedback.

1. Regarding the source of awareness: We acknowledge that the questionnaire did not capture how participants first learned about HPV (e.g., through the curriculum, peers, or media). This has now been explicitly clarified in the “Data collection tool” subsection of the Methods as follows:

“The questionnaire did not include an item assessing the source of HPV awareness (e.g., academic curriculum, peers, or media exposure).”

2. The Abstract has been updated to include key statistical values (e.g., proportions and p-values) to support the summarized findings.

---

## [Decision Letter · Decision Letter 1]

10 Nov 2025

Awareness, knowledge, and beliefs about Human Papillomavirus and its vaccine among Egyptian medical students: a cross-sectional national study

PONE-D-25-45022R1

Dear Dr. Baklola,

We’re pleased to inform you that your manuscript has been judged scientifically suitable for publication and will be formally accepted for publication once it meets all outstanding technical requirements.

Kind regards,

Nicola Serra

Academic Editor

PLOS ONE

Additional Editor Comments (optional):

Reviewers' comments:

Reviewer's Responses to Questions

**Comments to the Author**

Reviewer #1: (No Response)

Reviewer #2: All comments have been addressed

Reviewer #4: All comments have been addressed

2. Is the manuscript technically sound, and do the data support the conclusions?

Reviewer #1: (No Response)

Reviewer #2: Yes

Reviewer #4: Yes

3. Has the statistical analysis been performed appropriately and rigorously?

Reviewer #1: I Don't Know

Reviewer #2: I Don't Know

Reviewer #4: Yes

4. Have the authors made all data underlying the findings in their manuscript fully available?

Reviewer #1: Yes

Reviewer #2: Yes

Reviewer #4: Yes

5. Is the manuscript presented in an intelligible fashion and written in standard English?

Reviewer #1: Yes

Reviewer #2: Yes

Reviewer #4: Yes

Reviewer #1: The authors have correctly addressed all the requests, and the manuscript has been notably improved.

Reviewer #2: (No Response)

Reviewer #4: The authors have done a great job by addressing the comments raised in my previous submission. I am excited that the revised manuscript is now better and improved on the quality. The manuscript has demonstrated originality from the title/abstract section, introduction, methods, results and discussion sections are all well written to provide critical information needed on the field of public health

**Do you want your identity to be public for this peer review?** For information about this choice, including consent withdrawal, please see our Privacy Policy

Reviewer #1: No

Reviewer #2: No

Reviewer #4: **Yes: ** Francis Ajang Magaji

---

## [Editor Report · Acceptance letter]

PONE-D-25-45022R1

PLOS ONE

Dear Dr. Baklola,

I'm pleased to inform you that your manuscript has been deemed suitable for publication in PLOS ONE. Congratulations! Your manuscript is now being handed over to our production team.

Kind regards,

on behalf of

Dr. Nicola Serra

Academic Editor

PLOS ONE